# Cognitive Deficits in the Acute Phase of COVID-19: A Review and Meta-Analysis

**DOI:** 10.3390/jcm12030762

**Published:** 2023-01-18

**Authors:** Isabele Jacot de Alcântara, Anthony Nuber-Champier, Philippe Voruz, Alexandre Cionca, Frederic Assal, Julie A. Péron

**Affiliations:** 1Clinical and Experimental Neuropsychology Laboratory, Faculty of Psychology, University of Geneva, 1205 Geneva, Switzerland; 2Faculty of Medicine, University of Geneva, 1205 Geneva, Switzerland; 3Neurology Department, Geneva University Hospitals, 1205 Geneva, Switzerland

**Keywords:** COVID-19, cognition, global scales, review, meta-analysis

## Abstract

This meta-analysis was conducted to quantify the risk of patients exhibiting cognitive deficits in the acute phase of COVID-19 at the time of the first variants (i.e., before the vaccine) and quantify the potential vulnerability of older patients and those who experienced more severe respiratory symptoms. To this end, we searched the LitCovid and EMBASE platforms for articles, including preprints, and included all studies (*n* = 48) that featured a measurement of cognition, which encompassed 2233 cases of COVID-19. Of these, 28 studies reported scores on global cognitive efficiency scales administered in the acute phase of COVID-19 (up to 3 months after infection). We were able to perform a meta-analysis of proportions on 24 articles (N_patients_ = 943), and a logistic regression on 18 articles (N_patients_ = 518). The meta-analysis for proportion indicated that 52.31% of patients with COVID-19 exhibited cognitive deficits in the acute phase. This high percentage, however, has to be interpreted taking in consideration the fact that the majority of patients were hospitalized, and some presented neurological complications, such as encephalopathy. A bootstrap procedure with random resampling revealed that an age of 59 was the threshold at which one would be more prone to present cognitive deficits. However, the severity of respiratory symptoms did not influence the scores on a global cognitive efficiency scale. Overall, our results indicated that neuropsychological deficits were a major consequence of the acute phase of the first forms of COVID-19.

## 1. Introduction

It is now widely acknowledged that COVID-19 can lead to neurological complications such as impaired consciousness, acute cerebrovascular disease, headache, epilepsy, and peripheral nervous system manifestations [1]. This raises question about patients’ cognitive status in the acute phase of COVID-19, namely up to 3 months after infection, based on the World Health Organization (WHO)’s definition of the post-COVID-19 condition and its onset [2]. Several studies of neurological manifestations associated with the COVID-19 infection have mentioned the presence of cognitive impairment. For example, Rogers et al. [3] found that 8.2% of patients with COVID-19 displayed altered mental status, while Badenoch et al. [4] reported that 20.2% had objective cognitive impairment in the acute phase.

Regarding more specific impairments, Paterson et al. [5] described attentional and executive deficits associated with encephalopathy in patients following a COVID-19 infection. In most studies of the acute neurological consequences of COVID-19, cognitive functions have been presented purely descriptively. Several groups have nonetheless tried to establish patients’ cognitive profiles by conducting exhaustive neuropsychological assessments, using different neuropsychological tasks to evaluate functions such as attention, memory, and language. For instance, in their case report study, Whiteside et al. [6] described neuropsychological deficits (especially in memory and verbal fluency) in three patients with COVID-19. Nonetheless, given that neuropsychological assessments are often time consuming, and patients with COVID-19 have severe fatigue [7], it can be difficult to perform thorough neuropsychological testing in these patients, especially in the acute phase of the disease. As a result, few studies have involved exhaustive neuropsychological testing in the acute phase of COVID-19, and where this has been done, it has often concerned a very small sample. More studies have tried to measure patients’ cognitive functioning in the acute phase of COVID-19 with global cognitive efficiency scales. For instance, using either the mini-mental state examination (MMSE) or the Montreal Cognitive Assessment (MoCA), Alemanno et al. [8] showed that 80% of patients admitted to a COVID-19 rehabilitation unit had neuropsychological deficits. Even though global scales do not allow for a full understanding of patients’ cognitive status, they represent a first step towards better understanding of the potential cognitive complications of the infection. They can help researchers identify the profiles of patients who exhibit cognitive difficulties in the acute phase of COVID-19, as well as secondary variables that may favor cognitive complications, such as age and disease severity. Global scales also seem to have the advantage of being robust in the presence of variables such as fatigue [9].

From a neurological point of view, Rogers et al. [3] showed that reported neurological manifestations may differ according to the severity of respiratory symptoms following COVID-19 infection. Their meta-analysis revealed that headache, myalgia, anosmia and dysgeusia were more frequently reported by patients who did not require intensive care, whereas fatigue was reported regardless of illness severity. These results support the idea that neurological symptoms can differ according to the severity of the respiratory symptoms. Regarding cognitive symptoms, other studies among patients who required invasive respiratory treatment have shown that a stay in the intensive care unit (ICU) is often associated with impaired cognitive functions [10].

Concerning the variable of age, in their literature review, Gallo Marin et al. [11] found that being aged above 55 years was linked to more severe forms of COVID-19 and hence to potentially more severe cognitive deficits.

To our knowledge, no quantitative meta-analysis has so far been carried out to summarize the data on global cognitive efficiency in the acute phase of COVID-19 together with the potential impact of secondary variables on cognitive impairment. Most available reviews or meta-analyses highlight neurological or neuropsychiatric deficits, and pay scant attention to cognition. Furthermore, one of the few reviews of cognitive impairment following COVID-19 highlighted the fact that the most commonly used tools to assess these deficits are global scales, and that impairment ranges are highly heterogenous [12]. Hence, the present study was designed to (1) identify cognitive deficits in the acute phase of COVID-19 at the time of the first variants (i.e., before the vaccine), and (2) clarify the impact of age and severity of respiratory symptoms on the observed deficits, as these variables may partly explain the highly heterogenous proportions of cognitive impairment reported in available reviews. We formulated two hypotheses: (1) we expected to observe cognitive deficits in patients in the acute phase of COVID-19 [8]; (2) we predicted that older patients and those with more severe respiratory symptoms would display more cognitive deficits [10,11].

## 2. Materials and Methods

The method was designed around the PRISMA 2020 checklist [13].

### 2.1. Eligibility Criteria

For the present meta-analysis, we searched for all studies including preprints as they represent an important proportion of publications at the beginning of the pandemic, published between the beginning of the pandemic and March 2022, that tackled cognition in adults during the acute phase of COVID-19 (up to 3 months post-infection). In order to have a homogenous sample of COVID-19 patients, we only included studies that examined patients before the apparition of the vaccine (up to the beginning of 2021). To be included, studies had to report cognitive data and mention at least one type of positive COVID-19 test administered to their patients.

Owing to a dearth of studies featuring exhaustive cognitive assessments, we decided only to retain studies that used validated global scales. Studies that only mentioned neurological or psychological deficits, or which referred to cognitive deficits in a purely descriptive way, were excluded from the analyses. However, in order to provide a comprehensive overview of the literature on cognitive deficits in the acute phase of COVID-19, we retained articles that measured specific cognitive processes for the literature review. We excluded studies that examined the effects of quarantine and isolation on mental health and possible cognitive impairment. Studies that presented results for patients more than 3 months after infection were also excluded. Because only a small number of studies tackled cognitive deficits in the acute phase of COVID-19, there were no exclusion criteria related to comorbidities.

### 2.2. Information Sources

On 2 September 2021, we performed a literature search of LitCovid (https://www.ncbi.nlm.nih.gov/research/coronavirus/, accessed on 2 September 2021), a hub dedicated to publications in the field of COVID-19 that is connected to PubMed, and therefore encompasses the NCBI and NLM, as well as preprint servers such as Medrxiv. In order to update our search and extend it to a non-COVID-19 specialized platform, we also conducted a search of the EMBASE database, looking for items published between the start of the pandemic and 8 March 2022.

### 2.3. Search Strategy

As the LitCovid platform is exclusively dedicated to publications in the field of COVID-19, we used the keywords “neuropsychology, cognition”, in order to ensure a focused search. We also conducted a more precise search focused on the most widely used global scales, using the keywords “neuropsychology, MoCA, MMSE”. Regarding the EMBASE search, we used the search terms “neuropsychology OR cog* AND COVID-19 AND acute”.

### 2.4. Identification and Data Collection Process

The main criterion for identification was a cognitive assessment carried out during the acute phase of COVID-19. First, two independent reviewers (ANC and IJA) screened all the titles and abstracts of the articles found on the search engines. Both were involved in the selection process that was then confronted (Cohen’s kappa = 0.45; agreement = 90.52%) and disagreement was handled through discussion. Only articles that reached an agreement were retrieved. The identified articles were then split between the two reviewers and read in their entirety. Each reviewer extracted the data from half of the articles identified, focusing on cognitive data, sociodemographic data, type of hospitalization, comorbidities, psychiatric data, brain imaging, and results of confirmatory tests (either nucleic acid or serology) for COVID-19. All data included in the meta-analysis were double checked by the two reviewers.

### 2.5. Data

We included all articles that contained a measure of global cognition. Individual or mean raw scores on the global cognitive scale and the proportion of deficits were analyzed, along with information about age, sex, and severity of respiratory symptoms (no hospitalization = mild; hospitalization without invasive mechanical ventilation = intermediate; hospitalization with invasive mechanical ventilation = severe).

### 2.6. Statistical Analysis

#### 2.6.1. Proportional Meta-Analysis

We performed a proportional meta-analysis with all the studies that provided the proportion of deficits or the number of cognitively impaired patients and the size of the total sample. We used the MedCalc^®^ Statistical Software version 20.114 (MedCalc Software Ltd., Ostend, Belgium; https://www.medcalc.org, accessed on 5 September 2022) that uses a Freeman-Tukey transformation (arcsine square root transformation [14]), which can be efficient for stabilizing variance due to sample size differences [15,16], in order to calculate the weighted summary of proportion under a fixed and random effects model based on DerSimonian et al. [17] and compute 95% confidence intervals. Then, in order to identify the critical age above which the risk of presenting cognitive deficits in the acute phase was significantly increased, we performed a bootstrap procedure with random resampling, wherein the critical age was estimated by maximizing the difference between the proportion of deficits observed strictly below and equal or above a particular age.

#### 2.6.2. Logistic Regression for Severity

We conducted a logistic regression on JASP version 0.16.1.0 (JASP, Amsterdam, The Netherlands) to further explore the role of the severity of respiratory symptoms in the presence of cognitive impairment. In order to account for the potential influence of age we included this variable as a covariate. We included all studies that featured homogenous groups based on the severity of respiratory symptoms (at least 75% of the groups had the same level of severity), together with mean scores on a global cognitive efficiency scale.

### 2.7. Data Availability

The data used for the analysis are provided as Appendix A.

## 3. Results

### 3.1. Study Identification

The literature search yielded 2513 articles, 98 of which were identified by the two reviewers after screening the titles and abstracts. A total of 48 articles were included in the present study (see Figure 1). Of these, 28 studies were included in the meta-analysis. All articles were included in the literature review and the map illustrating the worldwide distribution, 24 in the proportion analysis, and 18 in the logistic regression.

### 3.2. Proportional Meta-Analysis

All studies providing the proportion of deficits or the number of cognitively impaired patients and the size of the total sample were included. Hence, 24 articles were identified for proportion analysis, encompassing 943 patients in the acute phase of COVID-19. The majority of these studies used either the MoCA or the MMSE to determine the presence of cognitive deficits. However, some studies used either different scales or variants of the MoCA or MMSE (e.g., a telephone version). Furthermore, not all studies used the same cut-off to determine the presence of cognitive impairment. To increase the amount of data, and also highlight the heterogeneity of the cognitive data available in the literature, we decided to keep these studies and to report the types of tests and cut-offs used (see Table 1). As the analysis revealed high heterogeneity among the studies (I2 = 92.98%; 95% CI [90.75, 94.67%]), we used the random effects to estimate the proportions and their weights. The model estimated that 52.31% (95% CI [39.66%, 64.81%]) of the patients in the acute phase of COVID-19 displayed cognitive deficits, as measured by a global efficiency scale (see Figure 2). Sample sizes ranged from 2 to 185 individuals, and the proportion of cognitive deficits from 2.7% to 100%. Regarding the critical age, the bootstrap approach estimated that the age of 59 was the critical age above which one would be more likely (at least twice the risk) to present cognitive deficits in 38.9% of the 1000 trials (Figure 3).

**Table 1 jcm-12-00762-t001:** Single group summary for proportions.

Article	Mean or Median Age (in Years) ± SD	Inclusion Criteria	Severity	Mean or Median Time (in Days) ± SD since Disease Onset	Scale Used	Cut-Off Used	Cognitive Deficits according to MoCA or MMSE (n/N)	Proportion (%)	95% CI Lower Limit	95% CI Upper Limit	Fixed Weight (%)	Random Weight (%)
Ermis et al. [18]	61 ± 13.3	Confirmed cases of COVID-19 that required hospitalization	Intermed.	Inpatients *	MoCA	26	8/13	61.54	31.58	86.14	1.45	4.05
Martillo et al. [19]	54 ± 12.9	Confirmed cases of COVID-19 that required ICU	Severe	30	MoCA	19	24/30	80	61.43	92.29	3.21	4.47
Heyns et al. [20]	na	Confirmed cases of COVID-19 that required hospitalization	Intermed.	Inpatients *	MoCA	26	21/38	55.26	38.30	71.38	4.03	4.55
Hosp et al. [21]	65 ± 14.4	Confirmed cases of COVID-19 with at least one neurological symptom	Intermed.	18.4 ± 2.3	MoCA	26	18/26	69.23	48.21	85.67	2.79	4.41
Pistarini et al. [22]	64 ± 11.9	Confirmed cases of COVID-19 admitted to a rehabilitation unit	Intermed.	Inpatients *	MoCA	26	29/40	72.5	56.11	85.40	4.24	4.56
Rousseau et al. [23]	62 (IQR 49–68)	Confirmed cases of COVID-19 that required ICU	Severe	94 (IQR 90–101)	MoCA	26	14/32	43.75	26.36	62.33	3.41	4.49
Patel et al. [24]	62 ± 15.7	Confirmed cases of COVID-19 admitted to a rehabilitation unit	Severe	Inpatients *	MoCA	26	62/77	80.51	69.91	88.66	8.07	4.71
Solaro et al. [25]	54 ± 4.8	Confirmed cases of COVID-19 with no neurological comorbidity and no delirium episode	Intermed.	Inpatients *	MoCA	23	13/32	40.62	23.70	59.35	3.41	4.49
Imamura et al. [26]	54 ± 13.3	Confirmed cases of COVID-19 admitted to rehabilitation post ICU	Severe	Inpatients *	MoCA	26	1/27	3.7	0.09	18.97	2.90	4.43
Jain et al. [27]	na	Confirmed cases of COVID-19 admitted to a rehabilitation unit	Intermed.	Inpatients *	MoCA	26	10/16	62.5	35.44	84.80	1.76	4.17
Monti et al. [28]	56 ± 10.5	Confirmed cases of COVID-19 that required ICU with at least one day of mechanical ventilation	Severe	61 (IQR 51–71)	MMSE ^&^	NI	1/37	2.7	0.07	14.16	3.93	4.54
Bayrak & Çadirci [29]	73 (IQR 65–90)	Confirmed cases of COVID-19 older than 64 years that required hospitalization	Intermed.	Inpatients *	MMSE	24	27/122	22.1	15.12	30.54	12.72	4.77
Tomasoni et al. [30]	na	Confirmed cases of COVID-19 that required hospitalization	Intermed.; Severe	~46 (IQR 43–48)	MMSE	18	10/21	47.6	25.71	70.22	2.28	4.32
Alemanno et al. [8]	67 ± 12.2	Confirmed cases of COVID-19 admitted to a rehabilitation unit	Intermed.; Severe	Inpatients *	MoCA/MMSE	NI	70/87	80.5	70.57	88.19	9.10	4.73
Raman et al. [31]	55 ± 13	Confirmed cases of COVID-19 that required hospitalization	Intermed.; Severe	69 (IQR 62–76)	MoCA	26	16/58	27.6	16.66	40.90	6.10	4.66
Kas et al. [32]	57 ± 9.2	Confirmed cases of COVID-19 with a related encephalopathy	Intermed.; Severe	Inpatients *	MMSE	24	2/2	100	15.81	100	0.31	2.48
Groiss et al. [33]	60 ± 20.4	Confirmed cases of COVID-19 that required mechanical ventilation	Severe	Inpatients *	MoCA/MMSE	26/24	4/4	100	39.76	100	0.52	3.09
Beaud et al. [34]	65 ± 7.6	Confirmed cases of COVID-19 that required mechanical ventilation	Severe	Inpatients *	MoCA	26	9/13	69.23	38.57	90.91	1.45	4.05
De Lorenzo et al. [35]	57 (IQR 48–67)	Confirmed cases of COVID-19	Mild; Intermed.; Severe	23 ^#^ (IQR 20–29)	MoCA	24	47/185	25.4	19.30	32.31	19.23	4.81
Negrini et al. [36]	60 ± 15.4	Confirmed cases of COVID-19 that required hospitalization with no neurological complications such as stroke	Intermed.; Severe	At least 30	MMSE	24	3/9	33.33	7.49	70.07	1.03	3.79
Delorme et al. [37]	69 ± 4.2	Confirmed cases of COVID-19 with a related encephalopathy	Intermed.; Severe	9 ± 3.5	MMSE	24	1/2	50	1.26	98.74	0.31	2.48
Di Pietro et al. [38]	60 ± 12.1	Confirmed cases of COVID-19 admitted to a rehabilitation unit	Severe	57 ± 20.7	MMSE	24	1/8	12.5	0.32	52.65	0.93	3.70
Jaywant et al. [39]	65 ± 13.9	Confirmed cases of COVID-19 admitted to a rehabilitation unit	Intermed.; Severe	43 ± 19.2	BMET	NI	46/57	80.7	68.09	89.95	6.00	4.65
Pirker-Kees et al. [40]	79 ± 8.4	Confirmed cases of COVID-19 that required hospitalization without history of neurological disease	Intermed.	15 ± 6.2	MoCA	23–25	6/7	85.7	42.13	99.64	0.83	3.59
Total (fixed effects)							443/943	46.51	43.33	49.71	100	100
Total (random effects)							443/943	52.31	39.66	64.81	100	100

^#^ Days after hospital discharge. * Mean time from disease onset not available, but assessment was performed in inpatient in the acute phase of the infection. ^&^ Telephone version. SD: standard deviation; MoCA: Montreal scale of Cognitive Assessment; MMSE: mini-mental state examination; BMET: Brief Memory and Executive Test; NI: not indicated; path.: pathological score; Mild: patients not hospitalized; Intermed.: patients in intermediate or intensive care without invasive ventilation; Severe: patients with invasive ventilation; na: not assessed.

### 3.3. Logistic Regression for Severity

We ran a logistic regression analysis dichotomizing the mean global cognitive scores (0 = unimpaired, 1 = impaired) of the groups of hospitalized patients who had or had not undergone invasive mechanical ventilation; we also added the variable of age as a covariate. Groups of non-hospitalized patients could not be added to the analysis, owing to a lack of relevant data. All studies that had a homogenous group of patients regarding the severity of respiratory symptoms and which provided a group mean score on a global cognitive efficiency scale were included (see Appendix A. The resulting logistic regression on 18 articles showed that the model including the severity variable and age explained 34% of the variance in global cognitive deficits (pseudo McFadden *R*^2^ = 0.34; *X*^2^ = 8.49, *p* = 0.01). However, none of these two variables were classified as a significant predictor (*z* = −1.50, *p* = 0.13; *z* = 1.44, *p* = 0.15).

## 4. Geographical Distribution of Articles

We created a world map to illustrate the worldwide distribution of the articles we identified. The majority of articles came from North America and Europe, with only a few coming from Asia, whilst none from Africa and Oceania (see Figure 4).

## 5. Literature Review

In the present literature review, we extracted central parameters concerning neurocognitive functioning from 48 articles. As mentioned earlier, we excluded papers that measured cognition more than 3 months after the onset of COVID-19. Most of the studies in our sample reported sociodemographic/clinical data, but there were wide disparities in the measures of cognition and brain imaging. In particular, we noted a considerable diversity of tools used for measuring memory and executive processes.

We found that 42 of the 48 articles measured cognition with global measurement scales: MMSE [41], MoCA [42], Functional Independence Measure (FIM [43]), Global Deterioration Scale (GDS [44]), Telephone Interview for Cognitive Status (TICS [45]), and Addenbrooke’s Cognitive Examination (ACE III [46]) (see Table 2). More specifically, for the measurement of complex cognitive processing, the tools used to measure executive functions (measured in 16 of the 48 articles) were also very varied, and probed different processes: Frontal Assessment Battery (FAB [47]), Trail Making Test (TMT [48]), Stroop [49], Symbol Digit Modalities Test (SDMT [50]), and the Brief Memory and Executive Test (BMET [51]) (see Table 2). For memory, we made the same observation (measures in 9 of the 48 articles), using the Hopkins Verbal Learning Test (HVLT [52]), California Verbal Learning Test (CVLT [53]), and Spain-Complutense Verbal Learning Test (TAVEC [54]) (see Table 2). The remaining aspects of cognition (e.g., praxis, visual perception, language) were seldom investigated, if at all (8 out of 48 articles). For the measurement of psychiatric symptoms, the data were just as abundant (23 out of 48 articles), with tests regarding the measurement of quality of life, sleep difficulties, anxiety, depressive symptoms, and other manifestations: Beck Depression Inventory (BDI [55]), State Trait Anxiety Inventory (STAI-Y [56]), Hamilton Rating Scale for Depression (HRSD [57]), SF-36 health survey (SF-36 [58]), Hospital Anxiety and Depression Scale (HADS [59]), Zung Self-Rating Depression Scale (ZSDS [60]), measure of health-related quality of life from the EuroQol Group (EQ-SD-3L [61]), Impact of Event Scale-Revised Form (IES-R [62]), GDS, General Anxiety Disorder-7 (GAD-7 [63]) and Depression, Anxiety and Stress Scale–21 Items (DASS-21 [64]) (see Table 2). Finally, we observed that only 15 of the 48 articles reporting measures of cognition and/or psychiatric disorders included imaging data. MRI was the most widely used modality (11/15), followed by CT (6/15), EEG (5/15), and PET (4/15) (see Table 2).

Table 2 indicates the presence or absence of psychiatric, neurological and cognitive measures in each of the 48 articles. It should be noted that methods or tools that were only used in a single study are not listed.

## 6. Discussion

The first aim of this meta-analysis was to determine whether cognitive deficits are an important outcome to consider in the acute phase of the first COVID-19 forms. These deficits, as measured by global cognitive efficiency scales, were found in 52.31% (95% CI [39.66%, 64.81%]) of patients on average. However, this high percentage has to be interpreted according to the population included in the study, namely a large majority of hospitalized patients, some of whom presented neurological complications such as encephalopathy that increased the likelihood of presenting cognitive deficits. Therefore, this high percentage may not be accurate for all the population affected by COVID-19, particularly mild forms that did not require hospitalization. Additionally, limitations, such as the use of global cognitive efficiency scales and the fact that they were designed to detect mild cognitive impairment in dementia [88] and are subject to ceiling effects [89], have to be considered. It is also important to note that not all studies provided or used the same cut-offs to determine the presence of cognitive deficits, which brought heterogeneity to the data. Nonetheless, our meta-analysis highlighted that cognitive deficits in the acute phase seem to be frequent, at least in the hospitalized sub-group of COVID-19 patients. Additionally, it shows considerable inconsistency in the information available in the literature. The proportion of cognitive deficits identified with a global cognitive efficiency scale ranged from 2.7% to 100%. This high heterogeneity could be explained by the use of variants in the scales, as well as large sample size differences, with lower proportions seen in larger samples or studies with adjusted versions of the global cognitive scales [28], and higher proportions seen in smaller samples [32,33]. Hence, our study allowed us to highlight these heterogeneities and to estimate the proportion of patients with cognitive deficits in the acute phase of COVID-19 based on a summary of the information available in the literature.

The second aim of this meta-analysis was to clarify the role of severity of respiratory symptoms and age in the occurrence of said cognitive deficits, and thereby explain some of the above-mentioned heterogeneity. Regarding respiratory severity, although these results must be interpreted in light of the limited power of our meta-regression, contrary to our initial hypothesis, our analysis showed that the severity of respiratory symptoms was not a significant predictor for the presence of impaired global cognitive efficiency scores. Interestingly, even if not expected, these results seem in line with other studies that have shown that despite the association between ICU and cognitive impairments, the severity of the respiratory disease, as evaluated by length of ICU stay and duration of mechanical ventilation, did not seem to have an influence [90]. According to other studies, levels of hypoxia and hypoxemia are better predictors of cognitive impairment [90,91]. A more recent review assessing postcritical illness cognition found that hypoxia, along with mechanical ventilation and length of ICU stay, were not associated with long-term cognitive impairment [92]. This review identified the presence of delirium in ICU as a much more relevant factor to consider, as it seemed to be more closely associated with persistent cognitive impairment [92]. Altogether, these results seem to indicate that the severity of respiratory symptoms is not the best predictor when assessing cognitive deficits in the acute phase of COVID-19. It is therefore important to consider the aforementioned variables, as well as other complications such as encephalopathy [93], as they could better predict cognitive deficits. Time from disease onset to the cognitive assessment is another aspect that may have biased our results, especially as slightly more studies assessed patients without invasive mechanical ventilation when they were still in hospital. Unfortunately, we could not include this as a covariate in our analysis, as not all studies provided the exact number of days between infection and cognitive assessment.

Regarding age, we proposed that older patients would exhibit more cognitive deficits, as these patients tend to develop more severe forms of COVID-19 [11]. Our bootstrap approach highlighted that an age of 59 was a critical threshold for an increased risk of cognitive deficits in the acute phase of a COVID-19 infection. These results would seem in line with Gallo Marin et al. [11] that indicated that an age above 55 was associated with more severe forms of COVID-19, as most hospitalized patients were older than 55. Additionally, studies assessing cognition in general post-ICU patients highlighted that age, as well as mechanical ventilation and its duration, were significant predictors of cognitive impairment [94]. Interestingly, our results regarding the influence of the severity of respiratory symptoms failed to highlight any link with cognition. However, additional analysis revealed that patients who required mechanical ventilation seemed to be younger than those who did not (see Appendix A), which would imply that the severity of respiratory symptoms would not be linearly associated with age. It can then be speculated that age could influence the presence of cognitive deficits independently of the severity of respiratory symptoms (i.e., the necessity of mechanical ventilation). This suggests that more attention should be drawn to hospitalized patients above 59, whether they require mechanical respiratory assistance or not.

Overall, even if our results did not confirm all our hypotheses, they are still relevant to neuropsychological research in COVID-19 for two main reasons. First, they confirm that objective cognitive deficits are an important outcome that should be investigated in this disease, as they seem to be relatively frequent and consistent with several hypotheses concerning the potential direct or indirect effects of the virus on the central nervous system [95]. Second, our results highlight the inconsistency of available data, starting with the heterogeneity of the description, assessment and interpretation of cognitive deficits. In the literature, cognitive deficit is a highly heterogeneous concept, ranging from self-reported cognitive difficulties to deficits revealed by exhaustive neuropsychological assessments. There is also considerable asymmetry in the nature of the studied data, as most of the articles we identified reported subjective cognitive complaints or the results of assessments with global scales, and very few studies featured more comprehensive cognitive assessments. A further issue with the consistency of the available cognitive data is that nearly all the articles reported single cases, mainly of men with severe COVID-19 who displayed cognitive deficits. Additionally, as we showed with our map (see Figure 4), studies of cognitive deficits resulting from COVID-19 have been mainly conducted in Europe and America, negatively impacting the representativeness of the cognitive data. Another problem with the available data is that some studies did not apply exclusion criteria such as premorbid comorbidities, and/or did not assess patients’ sensory abilities following infection (e.g., visual and auditory modifications [96,97]), which potentially influenced subsequent cognitive assessments. This limited our ability to hypothesize causal association between infection and cognitive deficits, and raises the possibility that some of the highlighted cognitive deficits were not due to the infection, or at least not entirely so. Moreover, as most of the analyzed studies did not have control groups, we cannot exclude the possibility that the reported cognitive deficits were due to other aspects, such as the pandemic situation. A study showed that 30% of individuals in quarantine/self-isolation reported cognitive impairment [98]. Even if this was not an objective assessment, this observation leaves open the possibility that general aspects of the pandemic had (and continue to have) cognitive consequences. Finally, it is important to consider that this study describes the state of the literature concerning cognitive status after a COVID-19 infection at the time of its first forms and before the availability of the vaccine, thus it is not clear if these results are consistent with more recent forms of COVID-19.

Nevertheless, our review highlights an imbalance between the need to assess cognitive deficits after onset of COVID-19 and the number of studies featuring the cognitive assessments required to confirm the proportion of cognitive deficits, as well as associated risk factors. New studies assessing cognition in the acute phase of COVID-19 have to consider the role of sociodemographic and clinical features, as well as other potential explanatory variables (e.g., delirium and encephalopathy), in order to confirm these results and improve current understanding of the prevalence of cognitive deficits and their causal relationship with COVID-19. A better comprehension of these aspects is essential for health and mental health services to be able to spot patients who are cognitively vulnerable and to offer improved and specific treatment.

## 7. Conclusions

Cognitive deficits seem to be a major consequence of COVID-19 in the acute phase. Our results suggest that the age of 59 is a potential threshold above which one would be more at risk of presenting cognitive deficits, and that the severity of respiratory symptoms did not seem to influence cognitive deficits. Moreover, this study shows the methodological limitations of global scales, highlighting the need to investigate cognition more thoroughly in the acute phase of COVID-19. Thus, it raises the importance of developing strict, exhaustive and standardized neuropsychological protocols to establish patients’ cognitive profiles and allow targeted neuropsychological rehabilitation to potentially improve the post-COVID-19 condition.

Abbreviations: Addenbrooke’s Cognitive Examination (ACE III); Beck Depression Inventory (BDI); Boston Naming Test (BNT); Brief Memory and Executive Test (BMET); California Verbal Learning Test (CVLT); confidence interval (CI); Depression, Anxiety and Stress Scale-21 Items (DASS-21); Frontal Assessment Battery (FAB); Functional Independence Measure (FIM); General Anxiety Disorder-7 (GAD-7); Global Deterioration Scale (GDS); Hamilton Rating Scale for Depression (HRSD); Hopkins Verbal Learning Test (HVLT); Hospital Anxiety and Depression Scale (HADS); Impact of Event Scale-Revised (IES-R); intensive care unit (ICU); Measure of health-related quality of life from the EuroQol Group (EQ-5D-3L); mini-mental state examination (MMSE); Montreal Cognitive Assessment (MoCA); SF-36 Health Survey (SF-36); Spain-Complutense Verbal Learning Test (TAVEC), State Trait Anxiety Inventory (STAI-Y); Symbol Digit Modalities Test (SDMT); Telephone Interview for Cognitive Status (TICS); Trail Making Test (TMT); Wechsler Memory Scale–Fourth Edition (WMS–IV); Zung Self-Rating Depression Scale (ZSDS).

## Figures and Tables

**Figure 1 jcm-12-00762-f001:**
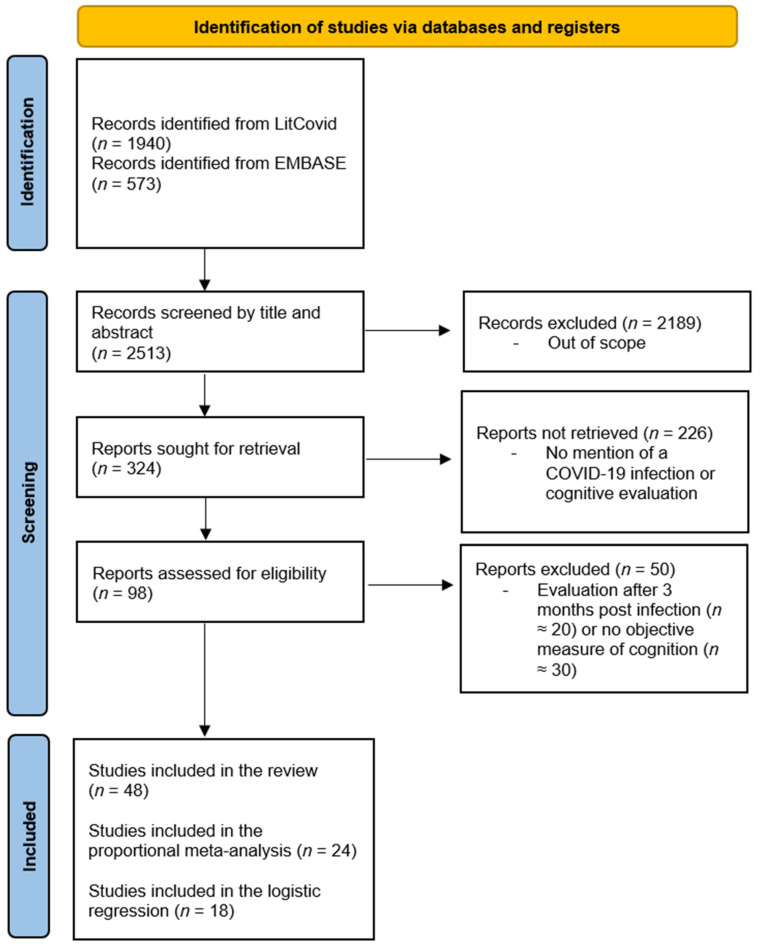
PRISMA flowchart of the article selection process over the two search periods. PRISMA = preferred reporting items for systematic reviews and meta-analyses.

**Figure 2 jcm-12-00762-f002:**
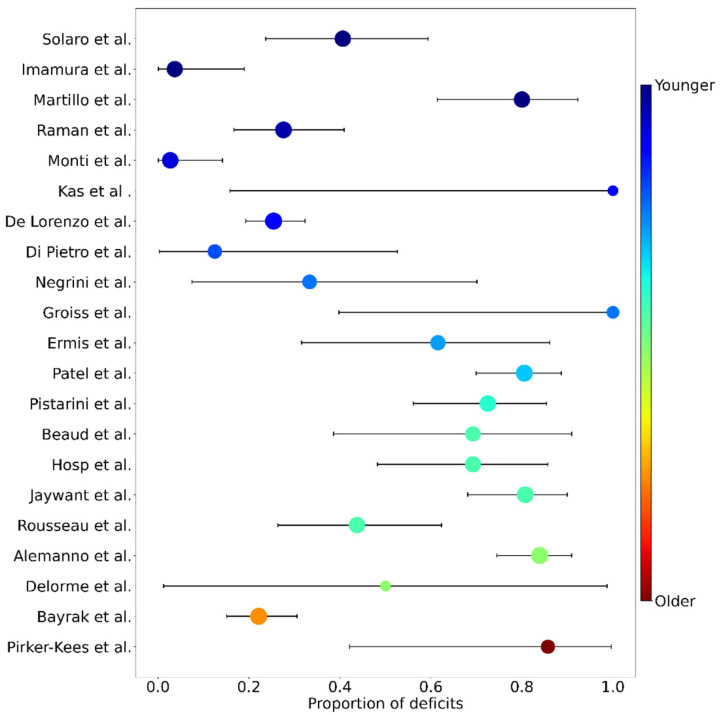
Forest plot of proportions of cognitive deficits for each study, as measured by global scales. Studies are sorted in increasing (from top to bottom) average age (from 54 to 78 years old). Dot sizes are proportional to the random effect weight, whiskers are the 95% confidence interval (both based on the sample size) and colors represent the average age (blue: younger; red: older). Studies are identified by their first author (y axis) and correspond to the following publications (from top to bottom): Solaro et al. [25]; Imamura et al. [26]; Martillo et al. [19]; Raman et al. [31]; Monti et al. [28]; Kas et al. [32]; De Lorenzo et al. [35]; Di Pietro et al. [38]; Negrini et al. [36]; Groiss et al. [33]; Ermis et al. [18]; Patel et al. [24]; Pistarini et al. [22]; Beaud et al. [34]; Hosp et al. [21]; Jaywant et al. [39]; Rousseau et al. [23]; Alemanno et al. [8]; Delorme et al. [37]; Bayrak & Cadirci [29]; Pirker-Kees et al. [40].

**Figure 3 jcm-12-00762-f003:**
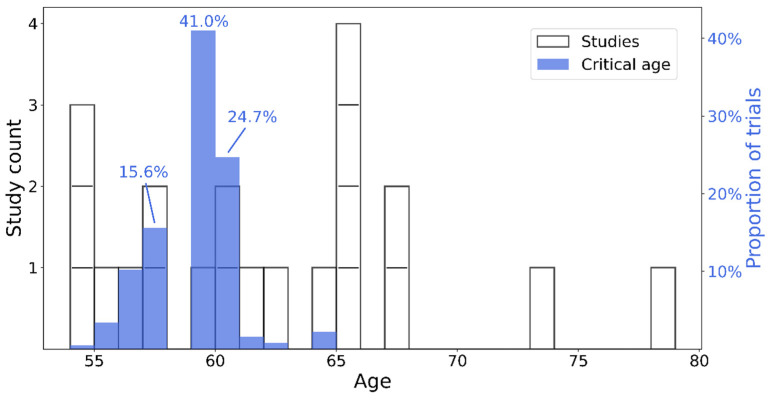
Histogram of age distribution of studies (white–left y axis) and critical age estimation (blue–right y axis). Individual studies are shown as white bar segments. The top three more frequent critical ages, as estimates in the bootstrap process, are highlighted by their respective percentage of positive identification throughout the 1000 trials.

**Figure 4 jcm-12-00762-f004:**
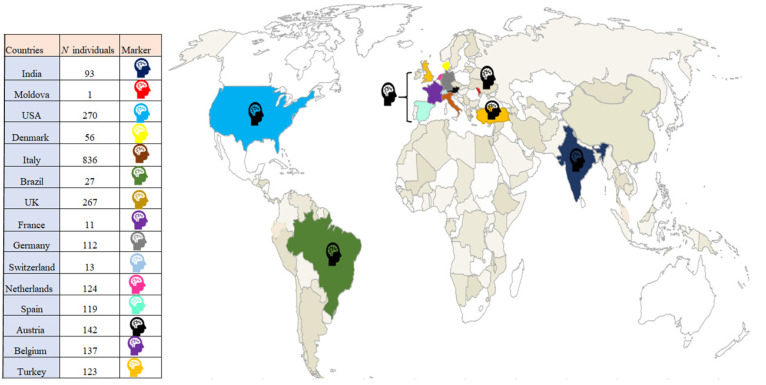
Distribution of articles in the meta-analysis and literature review that featured measures of cognition and presence of articles included (
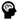
, color corresponds to geographical area).

**Table 2 jcm-12-00762-t002:** Parameters yielded by literature review.

Article	Demographic/Clinical				Cognition					Psychiatric		Imaging				Summary of Tests
	Age	Sex	Severity	Comorbidity	Global	Language	Executive	Memory	Other	Depression	Anxiety	MRI	PET	EEG	CT	
Amalakanti et al. [65]	√	√	√	√	√											MoCA
Ciolac et al. [66]	√	√			√					√						MoCA, BDI
Jaywant et al. [39]	√	√	√	√			√	√		√	√					BMET
Alemanno et al. [8]	√	√	√		√	√	√	√	√	√		√			√	HRSD, MMSE, MoCA
Raman et al. [31]	√	√	√	√	√					√	√	√				MoCA, GAD-7
Whiteside et al. [6]	√	√	√	√		√	√	√		√	√					Clinical examination
Kas et al. [32]	√	√		√	√		√						√	√		MMSE and clinical examination
Woo et al. [67]	√	√	√		√											TICS
Ortelli et al. [68]	√	√	√	√	√		√			√						MoCA, BDI, FAB
Groiss et al. [33]	√	√	√	√	√		√							√		MoCA, MMSE, SDMT
Van den Borst et al. [69]	√	√	√	√						√	√					HADS, TICS, CFQ
Beaud et al. [34]	√	√	√	√	√		√					√			√	MoCA, FAB
Vallecillo et al. [70]	√	√	√	√	√											Brief
Almeria et al. [71]	√	√	√	√	√	√	√	√		√	√					HADS, TAVEC, WMS-IV, TMT, SDMT, BNT
Chia et al. [72]	√	√	√	√	√		√	√				√				MoCA, FIM, FAM
De Lorenzo et al. [35]	√	√	√	√	√						√					STAI-Y, MoCA
Negrini et al. [36]	√	√	√	√	√					√	√					STAI, MMS
Priftis et al. [73]	√	√		√		√			√			√				Graphic and language subtests
Delorme et al. [37]	√	√		√	√		√					√	√	√	√	MMSE, FAB
Mcloughlin et al. [74]	√	√	√	√	√											TICS
Varatharaj et al. [75]	√	√		√												Clinical examination
Pinna et al. [76]	√	√	√	√	√			√								Clinical examination
Zambreanu et al. [77]	√	√	√	√	√							√				ACE III
Ermis et al. [18]	√	√	√	√	√							√		√	√	MoCA
Rass et al. [78]	√		√	√	√					√	√					SF-36, HADS, MoCA
Martillo et al. [19]	√	√	√	√	√					√	√					MoCA, EQ-5D-3L
Versace et al. [79]	√	√	√	√			√								√	FAB
Blazhenets et al. [80]	√	√	√		√								√			MoCA
Olezene et al. [81]	√	√	√	√	√											MoCA
Heyns et al. [20]	√		√		√					√	√					HADS, MoCA
Hosp et al. [21]	√	√	√	√	√		√	√				√	√			MoCA, TMT, HVLT
Yesilkaya et al. [82]	√	√	√		√		√	√				√		√	√	GDS, FAB, CVLT
Pirker-Kees et al. [40]	√	√	√		√											MoCA
Pistarini et al. [22]	√	√	√		√					√						MoCA
Di Pietro et al. [38]	√		√	√	√	√	√	√	√	√	√					MMSE, FAB, TMT, memory subtests
Rousseau et al. [23]	√	√	√	√	√					√	√					MoCA, EQ-5D-3L, HAD, IES-R
Patel et al. [24]	√	√	√	√	√											MoCA
Johnsen et al. [83]	√	√	√	√	√		√									SCIP, TMT
Bayrak et al. [29]	√	√	√	√	√					√						GDS, MMSE
Mazza et al. [84]	√	√	√	√	√					√	√					ZSDS
Solaro et al. [25]	√	√	√	√	√					√	√					HADS, MoCA
Imamura et al. [26]	√	√	√	√	√					√	√					DASS-21, MoCA, FIM
Udina et al. [85]	√	√	√	√	√		√									MoCA, SDMT
Greco et al. [86]	√	√		√	√											MMSE
Monti et al. [28]	√	√	√	√	√					√	√					HADS, MMSE, EQ-5D-3L
Peters et al. [87]	√	√	√	√	√							√				MoCA
Jain et al. [27]	√	√	√	√	√					√	√					MoCA
Tomasoni et al. [30]	√	√	√	√	√					√	√					MMSE, HADS

Note. Addenbrooke’s Cognitive Examination (ACE III); Beck Depression Inventory (BDI); Boston Naming Test (BNT); Brief Memory and Executive Test (BMET); California Verbal Learning Test (CVLT); Depression, Anxiety and Stress Scale–21 Items (DASS-21); Frontal Assessment Battery (FAB); Functional Independence Measure (FIM); General Anxiety Disorder-7 (GAD-7); Global Deterioration Scale (GDS); Hamilton Rating Scale for Depression (HRSD); Hopkins Verbal Learning Test (HVLT); Hospital Anxiety and Depression Scale (HADS); Impact of Event Scale-Revised (IES-R); Measure of health-related quality of life from the EuroQol Group (EQ-5D-3L); mini-mental state examination (MMSE); Montreal Cognitive Assessment (MoCA); SF-36 Health Survey (SF-36); Spain-Complutense Verbal Learning Test (TAVEC), State Trait Anxiety Inventory (STAI-Y); Symbol Digit Modalities Test (SDMT); Telephone Interview for Cognitive Status (TICS); Trail Making Test (TMT); Wechsler Memory Scale–Fourth Edition (WMS–IV); Zung Self-Rating Depression Scale (ZSDS).

## Data Availability

The data are available in Appendix A.

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
