# Peer review of "Cognitive Deficits in the Acute Phase of COVID-19: A Review and Meta-Analysis"

_jcm, 2023, doi:10.3390/jcm12030762_

Round 1
Reviewer 1 Report
Âlcantara et al. provide a literature review and meta-analysis to identify the proportion of COVID-19 patients exhibiting cognitive deficits at acute disease. They also analysed the contributions of age and disease severity as risk factors for cognitive disturbance in COVID-19. This is an important topic; as the authors highlight, although it is clear that COVID-19 induces neurological disturbance, the proportion of patients at risk of developing cognitive symptoms varies largely in different reports. However, some major methodological issues preclude the interpretation of the authors’ main finding (that roughly half of the COVID-19 patients present cognitive deficits in the acute phase of the disease).
Major issues
- In the flowchart from figure 1, authors should include the specific reasons why articles were excluded from the analysis and the number of articles excluded for each reason. Why were so many reports not retrieved? Did authors only include those with open access?
The flowchart should also include the number of articles and the total n of patients included for each analysis performed (i.e. proportional meta-analysis and logistic regression).
- The proportional meta-analysis included studies with biased patient inclusion criteria leading to a biased conclusion. For example, the study from Kas et al. included only COVID-19 patients presenting encephalopathy, which resulted in a 100% estimated proportion of cognitive deficits - this is a proportion that reflects this study-specific inclusion criterion and that does not reflect the general population infected with COVID-19. As such, the authors’ main conclusion that 52.3% of COVID-19 patients exhibited cognitive deficits in the acute phase is equally biased. The authors should exclude studies with biased inclusion criteria from proportional meta-analysis and prevent themselves from providing such a misleading conclusion.
- The authors state in the discussion that mechanical ventilation was more frequent in younger patients in their population sample. This disagrees with the literature and, perhaps, shows that the population analysed does not reflect the population of COVID-19-infected patients. Further, it should be clearly shown in the results session as an important finding.
Minor issues
- Concerning the logistic regression for COVID-19 severity analysis:
a. Were patients that died from COVID excluded from the analysis?
b. Invasive mechanical ventilation requires sedation. If cognitive assessment happened before mechanical ventilation, was the patient considered a severe or intermediate case?
c. Why were non-hospitalised patients excluded from the analysis? Shouldn’t they be included as mild cases?
- Concerning fig. 4:
a. It is named Fig.5 on the results text (pag.10, last paragraph).
b. It needs to be clarified what the marker means.
c. It would be more informative if it contained the n of patients, not the number of studies.
- Table 3’s headline is not readable.
- Why was the evaluated period set differently in the two search databases?
Reviewer 2 Report
The results are relevant to neuropsychological research in COVID-19 because they confirm that objective cognitive deficits are an important outcome that should be investigated in these patients.
I have some methodological questions that the authors should address before publication.
The authors stated that they searched for all the studies, including preprints, published between the beginning of the pandemic and the end of April 2022. However, they mention that to update their search and extend it to a non-COVID-19 specialized platform, they also conducted a search of the EMBASE database, looking for papers published between the start of the pandemic and 8 March 2022. I doubt the exact period of the search (march or April 2022). As we are in December 2022, I suggest they update the search.
Additionally, I believe that some relevant publications were missing. For instance, Tolentino et al. (available online in 2020- psychiatry and clinical neurosciences) described an extensive neuropsychological examination in a COVID-19 patient (doi: 10.1111/pcn.13178). It should be mentioned that the authors of the present review mentioned that neuropsychological assessments are often time-consuming, patients with COVID-19 have severe fatigue, and it is difficult to perform thorough testing in these patients. However, do Carmo et al. (Journal of Psychiatric Research 150, 189-196, available online in 2021) published a study in COVID patients using a brief assessment.
They applied software that uses a Freeman-Tukey transformation (arcsine square root transformation to calculate the weighted summary of proportion under a fixed and random effects model. I believe that this choice should be better explained in the revised manuscript.
Round 2
Reviewer 1 Report
I thank the authors for providing a letter with responses point by point.
However, the potential biased proportional analysis remains a major issue. Although the authors have included a few sentences in the discussion, most readers may oversee it. Therefore, a similar statement should be included in the abstract.
Additionally, it would increase the transparency of the analysis if authors included a column in Table 1 to describe the major patient inclusion criteria from the included studies (if it was a broad or specific COVID-19 population studied).
Reviewer 2 Report
I thank the authors for their reply. However, some points still need a better explanation.
Abstract
According to them, LitCovid and EMBASE platforms for articles were searched for cognitive changes in the acute phase of COVID-19. They included preprints. Therefore, the inclusion of preprints should be mentioned.
The literature search consisted of 48 publications that featured a cognition measurement encompassing 2233 cases of COVID-19. Of these, 28 studies reported scores on global cognitive efficiency scales administered in the acute phase of COVID- 19 (up to 3 months after infection). However, in the abstract, It should be stressed that the total number of subjects (metanalysis) was 943.
Introduction:
The rationale for including papers that provided performances in global cognition still needs a better explanation. According to the authors, neuropsychological assessments are often time-consuming, and patients with COVID-19 have severe fatigue. However, several papers have described cognitive changes in the acute phase of COVID-19 that applied brief tests (for instance, SDMT and CVAT take only 90 seconds).
Alemanno et al. described cognitive deficits in COVID-19 patients after respiratory assistance in the subacute phase. Does “subacute” differ from “acute”?
Methods.
The authors explained that their search could not find those that did not contain their keywords. Although they restricted their quantitative analyses to papers using global cognitive changes (MMSE or MOCA,), it is clear that the systematic review of the literature (qualitative) deserves a more careful search. Even a simple search in google would let them find several other papers, including reviews on cognitive changes in the acute phase of COVID-19.
According to them, case reports were not included (for example, Tolentino et al.). It should be clearly stated in the methods section. This isn't very clear because there is an excellent description of a case report in the introduction (Whiteside et al.).
The inclusion of preprints should be justified.
Results
Table three provides a view of the literature search. I liked this broad approach (at least three case reports were included - Almufarrij et al., Chia et al., Zamobranu et al.- ). Table 3 also included papers that do not use global cognitive measurements. The authors should revise to include other papers on specific cognitive changes in the acute phase of COVID-19.
In the quantitative analysis, the study from Kas et al. and Delorme et al. included only two COVID-19 patients.
The cut-off for MMSE varied from 18 to 26. One hundred eighty-one patients do not have cut-offs (181/943= 19%).
· Discussion
A discussion about the difficulties associated with using different cut-offs should be included.
